# Individual differences in selective attention predict speech identification at a cocktail party

**Daniel Oberfeld\*, Felicitas Klöckner-Nowotny**

Department of Psychology, Section Experimental Psychology, Johannes Gutenberg-Universität, Mainz, Germany

**Abstract** Listeners with normal hearing show considerable individual differences in speech understanding when competing speakers are present, as in a crowded restaurant. Here, we show that one source of this variance are individual differences in the ability to focus selective attention on a target stimulus in the presence of distractors. In 50 young normal-hearing listeners, the performance in tasks measuring auditory and visual selective attention was associated with sentence identification in the presence of spatially separated competing speakers. Together, the measures of selective attention explained a similar proportion of variance as the binaural sensitivity for the acoustic temporal fine structure. Working memory span, age, and audiometric thresholds showed no significant association with speech understanding. These results suggest that a reduced ability to focus attention on a target is one reason why some listeners with normal hearing sensitivity have difficulty communicating in situations with background noise.

\*For correspondence: oberfeld@ uni-mainz.de

**Competing interests:** The authors declare that no competing interests exist.

## Introduction

Imagine yourself sitting at a table in a crowded restaurant, chatting with a friend of yours. Can you follow the conversation with your friend easily, or does the high noise level in general or more specifically the conversations heard from other tables interfere with speech intelligibility? While the human auditory system has impressive abilities in structuring the mixture of sound waves arriving at the ears into different auditory objects or streams (auditory scene analysis; e.g., *Bregman, 1990*; *Carlyon, 2004*), listeners show considerable variation when it comes to speech understanding in adverse acoustic conditions, such as the almost proverbial '*cocktail-party*' situation described above (*Bronkhorst, 2000*; *Cherry, 1953*). Surprisingly, pronounced individual differences in speech identification in background noise are observed even in listeners with *normal hearing*, that is, with audiometric thresholds better than 20 dB HL within the frequency range most important for speech (100 Hz–4 kHz; *Byrne et al., 1994*), usually taken as an indication of approximately intact processing in the inner ear. For example, *Ruggles and Shinn-Cunningham (2011)* tested normal-hearing subjects in a simulated cocktail-party listening task with two competing speakers presented 15° to the left and right of the target speaker. Across listeners, the percentage of correct responses in reporting a sequence of digits produced by the target speaker varied between 40% and 85% in an anechoic condition. Compatible with these experimental results, in clinical settings a relevant number of patients with normal audiometric findings complain about hearing difficulties in daily life (*Zhao and Stephens, 2007*).

Understanding speech in a cocktail-party situation with interfering speakers and other background noise requires *selective attention* (*Bronkhorst, 2015*; *Cherry, 1953*; *Shinn-Cunningham, 2008*; *Xiang et al., 2010*), and was even proposed to be the '*best-known real life example of selective attention*' (*Pashler, 1998*). The information from the target speaker needs to be processed,

while information from other sound sources should be ignored. The main hypothesis tested in the present study was that individual differences in the *capability to direct auditory selective attention to the relevant stimulus in the presence of distractors* explain a significant proportion of the inter-individual variance in cocktail-party listening performance. Although several aspects of speech appear to be processed outside the focus of attention (*Pulvermüller and Shtyrov, 2006*), attention enhances the representation of speech (and other sounds) at relatively early stages (e.g., *Choi et al., 2014*; *Srinivasan et al., 2012*; *Xiang et al., 2010*; *Zion Golumbic et al., 2013*). On the behavioral level, the importance of attention is illustrated by studies that manipulated the a-priori information concerning the target speaker. For instance, in an experiment by *Kidd et al. (2005)*; the speech identification performance was better when the listener knew in advance which of three talkers (presented at different spatial positions) would be the target speaker, compared to conditions where the target location was uncertain. Thus, for exactly identical acoustic signals, being able to direct attention to the correct location results in a large improvement in speech recognition (e.g., *Best et al., 2007*; *Kitterick et al., 2010*). The direction of selective attention to the target speaker can be impaired due to limitations imposed by the *acoustic signal* or by *perceptual or cognitive characteristics of the listener* (cf. *Mattys et al., 2012*). If a listener has problems in using acoustic cues for the formation of auditory objects or streams, or on a more cognitive level is incapable of ignoring irrelevant information, then speech identification performance will be low even when fundamental frequency, timbre, and spatial location of the speakers differ.

With respect to listeners' attentional capabilities, it is important to take into consideration that attention is a multifaceted phenomenon (cf. *Styles, 2006*). In a cocktail party situation, it is required to attend to a certain speaker (target) and to ignore the other sound sources (distractors). We were interested in whether speech understanding in a cocktail-party situation could be related to a more general ability to focus attention on a target in the presence of distractors. Surprisingly, this particular aspect of attention has not been studied very systematically in previous experiments that investigated the role of attention for speech understanding in noise (*Füllgrabe et al., 2014*; *Gatehouse and Akeroyd, 2008*; *Heinrich et al., 2015*; *Neher et al., 2009*, *2011*, *2012*; *Schoof and Rosen, 2014*; *van Rooij et al., 1989*). Most of these studies used tests developed for neuropsychological settings, like the Test of Everyday Attention (TEA; *Robertson et al., 1996*) and the Trail Making Tests (TMT; *Bowie and Harvey, 2006*), indexing *visual search* (TEA Map Search, Telephone Search), *task/attentional switching* (TEA Visual Elevator and Auditory Elevator with Reversal, TMT-B), *sustained attention* (TMT-A, TEA Lottery Test and Elevator Counting), or *divided attention* (TEA Telephone Search While Counting). Only one TEA subtest ("Elevator counting with distraction") directly addresses the capability to ignore distractors. Here, participants have to count low pitch tones ('targets') while ignoring interspersed high pitch tones ('distractors'). We are aware of only two studies that included this subtest (*Gatehouse and Akeroyd, 2008*; *Neher et al., 2009*), in hearing-impaired listeners. Concerning the other tasks, while switching attention is relevant for situations where the target speaker changes dynamically, for example in a conversation involving more than two persons, there are many situations where the target speaker does not change. Visual search and sustained attention seem even less relevant for cocktail-party listening. For this reason, our study included tasks in which subjects had to *identify an auditory or visual target element in the presence of distractors*. In our view, this is the most important aspect of attention in cocktail-party listening.

As a measure of *visual selective attention*, we used a *flanker task* as established by *Eriksen and Eriksen (1974)*; where a target stimulus is surrounded by task-irrelevant distractors (flankers). In the critical *incompatible condition*, the flankers and the target call for opposite responses. If the incompatible flankers produce only small response time (RT) costs, then the participant has a high ability to focus visual selective attention on the target stimulus. This flanker interference is defined as the difference between the average response time (RT) in the incompatible condition and in a neutral condition where the flankers are not associated with one of the responses relevant for the target.

To measure the individual ability to direct *auditory selective attention* to a target stimulus while ignoring distractors, we used an intensity discrimination task under backward masking. If – as in the present study – a target sound is followed by a backward masker after a silent inter-stimulus interval (ISI) of 50 ms or more, it is virtually impossible that the masker affects the representation of the target in the auditory nerve (*Kiang et al., 1965*; *Plack and Viemeister, 1992*). Instead, strong effects of the backward masker on intensity discrimination can be explained by a failure to selectively attend

to the target sounds while ignoring the maskers (*Oberfeld and Stahn, 2012*; *Oberfeld et al., 2012*; *Schlauch et al., 1997*). For example, in a study from our lab that quantified the amount of attention directed to the maskers using a behavioral reverse-correlation approach (*Oberfeld et al., 2014*), the effect of non-simultaneous masking on the intensity difference limen (DL) was well accounted for by the attention to the maskers, explaining 72% of the variance.

Listeners with normal audiometric thresholds may differ in their sensitivity to the temporal fine structure (TFS) of sounds (e.g., *Füllgrabe, 2013*; *Ruggles et al., 2011*), which is necessary for using interaural time difference (ITD) cues to sound localization. These differences were proposed to be due to cochlear neuropathy, which could for instance be caused by moderate noise exposure (*Bharadwaj et al., 2015*; *Kujawa and Liberman, 2009*) and is sometimes described as 'hidden hearing loss' (*Plack et al., 2014*) because it cannot be detected using standard measures of audiometric threshold. Several studies showed a correlation between TFS sensitivity and the recognition of speech in noise, for normal-hearing as well as for hearing-impaired listeners (*Bharadwaj et al., 2015*; *Füllgrabe et al., 2014*; *Neher et al., 2011*, *2012*; *Ruggles et al., 2011*; *Schoof and Rosen, 2014*). For this reason, our study included binaural sensitivity for the temporal fine structure as a potential predictor of speech identification in a cocktail-party situation, using a task proposed by *Hopkins and Moore (2010)* that measures the smallest detectable interaural phase difference (IPD) of a sinusoidal carrier relative to an IPD of 0°.

As additional cognitive measures, working memory capacity (e.g., *Akeroyd, 2008*; *Füllgrabe and Rosen, 2016*) measured in a sentence span test (*Daneman and Carpenter, 1980*), and processing speed (e.g., *Salthouse, 1996*; *Tun and Wingfield, 1999*) measured by the RT in the neutral condition of the visual flanker task, were included as potential predictors of speech-in-noise identification. The latter was measured in a simulated cocktail-party listening situation with two competing speakers that were presented 25° to the left and right of the target speaker, who was positioned in front of the listener (azimuthal angle 0°). In addition, self-reported hearing-related problems in daily life were assessed via the Speech, Spatial and Qualities of Hearing Scale (SSQ) by *Gatehouse and Noble (2004)*, using the German version (*Kießling et al., 2011*).

## Results

To which extent did speech understanding in a cocktail-party situation depend on the capability of directing selective attention to a target in the presence of distractors, binaural sensitivity for the temporal fine structure (TFS), and other factors? To answer this question, a multiple linear regression analysis was conducted (the statistical details are described in Materials and methods). The criterion variable was the speech recognition score (SRS) defined as the proportion correct in the simulated cocktail-party listening task with two interfering speakers. We used a sentence identification task based on a German matrix test (*Wagener et al., 1999a*) and presented binaural simulations of an anechoic environment (see Materials and methods). The nine predictors were (1) the elevation of the intensity difference limen caused by the backward masker ($DL_{elev}$), defined as the difference between the DL under masking and the DL in quiet, which measures the capability of directing auditory selective attention to a target (*Oberfeld et al., 2014*), (2) the amount of flanker interference in the flanker task ($Int_{Flanker}$), which indexes visual selective attention, (3) the IPD threshold in the TFS-LF (*Hopkins and Moore, 2010*) test ($TFS_{th}$), which measures binaural sensitivity for the temporal fine structure, (4) the pure-tone average threshold on the better ear ($PTA_{BE}$) at octave frequencies between 125 Hz and 4 kHz, (5) the average asymmetry in the hearing thresholds between left and right ear in the same frequency range ($HL_{diff}$), (6) the intensity-DL in quiet ($DL_{quiet}$), which represents a suprathreshold measure of hearing ability that is not related to selective attention, (7) the response time in the neutral condition of the flanker task ($RT_{neutral}$), which was included as a measure of processing speed (*Salthouse, 2000*), and (8) the proportion of correctly recalled consonants in the sentence span task ($SS_{Pcorr}$) that indexes working memory capacity. Finally, (9) the age of the participant was added as a predictor, as in previous studies (e.g., *Neher et al., 2012*), to investigate whether the observed inter-individual differences in cocktail-party listening are determined by other factors related to age. Note that due to the relatively large sample size it was not necessary to summarize the different predictors into a small number of factors as in some previous studies (*Füllgrabe et al., 2014*; *Heinrich et al., 2015*; *Schoof and Rosen, 2014*; *van Rooij et al., 1989*).

**Table 1.** Results of the multiple regression analysis. Criterion variable: speech recognition score (SRS; proportion correct) in the simulated cocktail-party listening task. Predictors: age, masker-induced elevation of the intensity difference limen ($DL_{elev}$), the amount of flanker interference in the flanker task ($Int_{Flanker}$), IPD threshold in the TFS-LF task ($TFS_{th}$), pure-tone average thresholds on the better ear ($PTA_{BE}$), average asymmetry in the hearing thresholds between left and right ear ($HL_{diff}$), intensity-DL in quiet ($DL_{quiet}$), response time in the neutral condition of the flanker task ($RT_{neutral}$), and proportion of correctly recalled consonants in the working memory task ($SS_{Pcorr}$). All variables were $z$-standardized.

| Predictor | β | SE | t | p | GDW | $β_{Lasso}$ |
|---|---|---|---|---|---|---|
| Intercept | 0.068 | 0.096 | 0.710 | 0.480 | | 0.081 |
| Age | 0.194 | 0.110 | 1.760 | 0.086 | 0.020 | − |
| **$DL_{elev}$** | **−0.347** | **0.107** | **3.240** | **0.003** | **0.152** | −0.220 |
| **$Int_{Flanker}$** | **−0.233** | **0.103** | **2.270** | **0.029** | **0.052** | −0.081 |
| **$TFS_{th}$** | **−0.383** | **0.103** | **3.730** | **0.001** | **0.204** | −0.286 |
| $PTA_{BE}$ | 0.137 | 0.102 | 1.350 | 0.186 | 0.016 | − |
| $HL_{diff}$ | −0.088 | 0.106 | 0.830 | 0.413 | 0.007 | − |
| $DL_{quiet}$ | −0.007 | 0.114 | 0.070 | 0.948 | 0.021 | |
| $RT_{neutral}$ | −0.037 | 0.129 | 0.280 | 0.778 | 0.015 | − |
| $SS_{Pcorr}$ | 0.193 | 0.111 | 1.740 | 0.091 | 0.085 | 0.089 |
| | | | | | $R^2 =0.57$ p<0.001 | $R^2 =0.44$ |

Note: $N = 45$. β: estimated ordinary least-squares (OLS) regression coefficient. SE: standard error of the estimate. t: t-statistic. Bold font indicates a β significantly different from 0 (p<0.05). GDW: general dominance weight. $β_{Lasso}$: regression coefficients for predictors selected by the Lasso procedure (model selection via four-fold cross-validation).

The regression model showed a good fit, $R^2 = 0.57$, p<0.001, $N = 45$. As can be seen in *Table 1*, the performance in the cocktail-party listening task was significantly negatively related to the intensity-DL elevation under backward masking. Thus, compatible with our hypotheses, participants who showed a better capability of focusing attention on the target sounds in the intensity discrimination task were less affected by the interfering speakers in the cocktail-party listening task. In the same line of reasoning, the significant negative regression coefficient for flanker interference shows that a high capability of directing visual selective attention corresponded to good performance on the cocktail-party listening task. The IPD threshold measured in the TFS-LF task was also significantly negatively related to the SRS. Thus, compatible with previous studies (e.g., *Füllgrabe et al., 2014*; *Neher et al., 2011*, *2012*; *Ruggles et al., 2012*), listeners who showed high sensitivity for the TFS performed better in the spatial listening task. None of the remaining predictors showed a significant association with the performance in the spatial listening task. Notably, neither for age nor for working memory capacity did the regression coefficient differ significantly from 0.

What can be concluded about the relative importance of the different psychoacoustic and cognitive predictors for explaining individual differences in cocktail-party listening? In our data, the nine predictors were partly correlated (see *Table 2*). In such a case, it can be misleading to gauge the relative importance of the predictors by considering the squared standardized regression coefficients (cf. *Tonidandel and LeBreton, 2011*). For this reason, we used the 'dominance analysis' approach proposed by *Budescu (1993)*; which was shown to be a useful measure of the relative importance of predictors in a regression model, both on theoretical grounds and in simulation studies (*LeBreton et al., 2004*; *Thomas et al., 2014*; *Tonidandel and LeBreton, 2011*). Dominance analysis provides a quantitative measure of relative importance by examining the change in the variance-accounted-for ($ΔR^2$) resulting from adding a predictor to all possible regression models containing subsets of the predictors. For example, if there are three predictors (A, B, and C), then there are four possible subset models to which predictor C can be added (that is, models containing only the

**Table 2.** Pairwise Pearson partial correlation coefficients, controlling for age. N = 50. In each row, the upper numbers are the partial correlation coefficients ($\rho_{partial}$), and the lower numbers are the p-values for the test of $|\rho_{partial}| > 0$. The rightmost column shows Pearson correlation coefficients with age. Bold font: p<0.05. Italics: p<0.10.

| | DL$_{elev}$ | Int$_{Flanker}$ | TFS$_{th}$ | PTA$_{BE}$ | HL$_{diff}$ | DL$_{quiet}$ | RT$_{neutral}$ | SS$_{Pcorr}$ | SSQ$_{speech}$ | SSQ$_{spatial}$ | SSQ$_{qualities}$ | Age |
|---|---|---|---|---|---|---|---|---|---|---|---|---|
| OLSA$_{Pcorr}$ | −0.374 | −0.149 | −0.353 | 0.060 | −0.163 | −0.244 | −0.232 | 0.338 | 0.121 | 0.083 | 0.230 | 0.033 |
| | 0.008 | 0.307 | 0.013 | 0.683 | 0.263 | 0.091 | 0.109 | 0.018 | 0.407 | 0.570 | 0.112 | 0.819 |
| DL$_{elev}$ | | −0.047 | 0.038 | 0.030 | −0.083 | −0.045 | 0.092 | −0.255 | −0.302 | −0.049 | −0.178 | −0.045 |
| | | 0.748 | 0.793 | 0.838 | 0.570 | 0.759 | 0.529 | 0.077 | 0.035 | 0.740 | 0.220 | 0.754 |
| Int$_{Flanker}$ | | | −0.081 | −0.018 | −0.234 | −0.032 | −0.094 | −0.112 | −0.141 | −0.098 | −0.112 | 0.045 |
| | | | 0.578 | 0.903 | 0.105 | 0.826 | 0.522 | 0.444 | 0.335 | 0.501 | 0.444 | 0.758 |
| TFS$_{th}$ | | | | 0.034 | −0.023 | 0.399 | 0.312 | −0.177 | −0.149 | −0.314 | −0.352 | 0.027 |
| | | | | 0.818 | 0.873 | 0.005 | 0.029 | 0.224 | 0.306 | 0.028 | 0.013 | 0.852 |
| PTA$_{BE}$ | | | | | −0.292 | −0.083 | −0.010 | −0.092 | 0.136 | 0.082 | 0.097 | −0.248 |
| | | | | | 0.042 | 0.572 | 0.944 | 0.531 | 0.353 | 0.577 | 0.508 | 0.082 |
| HL$_{diff}$ | | | | | | 0.195 | 0.038 | 0.087 | 0.274 | 0.198 | 0.119 | 0.227 |
| | | | | | | 0.180 | 0.795 | 0.551 | 0.057 | 0.174 | 0.416 | 0.113 |
| DL$_{quiet}$ | | | | | | | 0.383 | −0.115 | −0.008 | −0.118 | −0.130 | 0.082 |
| | | | | | | | 0.007 | 0.431 | 0.959 | 0.420 | 0.375 | 0.573 |
| RT$_{neutral}$ | | | | | | | | −0.198 | −0.087 | −0.135 | −0.011 | 0.217 |
| | | | | | | | | 0.172 | 0.552 | 0.355 | 0.940 | 0.129 |
| SS$_{Pcorr}$ | | | | | | | | | −0.011 | 0.081 | 0.053 | −0.380 |
| | | | | | | | | | 0.940 | 0.580 | 0.720 | 0.006 |
| SSQ$_{speech}$ | | | | | | | | | | 0.707 | 0.728 | 0.040 |
| | | | | | | | | | | <0.0001 | <0.0001 | 0.784 |
| SSQ$_{spatial}$ | | | | | | | | | | | 0.701 | 0.123 |
| | | | | | | | | | | | <0.0001 | 0.393 |
| SSQ$_{qualities}$ | | | | | | | | | | | | 0.058 |
| | | | | | | | | | | | | 0.692 |

intercept term, intercept and predictor A, intercept and predictor B, and intercept and predictors A and B, respectively). A predictor's *general dominance weight* (GDW; **Azen and Budescu, 2003**) is found by averaging the squared semipartial correlations across all of the possible subset models. This measure indexes a variable's contribution to the prediction of the dependent variable, by itself and in combination with the other predictors. The sum of the GDWs is the total proportion of variance explained by the regression model, $R^2$.

As shown in *Table 1*, the general dominance weight was highest for the IPD threshold in the TFS-LF task, followed by the DL-elevation in the intensity discrimination task. According to these results, sensitivity for the binaural TFS and auditory selective attention are the most important predictors of cocktail-party listening. The contribution of flanker interference was much lower, and the GDW for the non-significant predictor sentence span was even slightly higher than the GDW for flanker interference.

To validate the conclusions based on dominance analysis, we used a second approach to variable selection. In the *Lasso* method proposed by **Tibshirani (1996)**; regression coefficients for predictors with only a small explanatory value are set to 0 (*shrinkage*), so that the Lasso method effectively performs *subset selection*, that is, selects the most important predictors (**James et al., 2013**). The Lasso approach is widely used in the field of statistical learning (**Hastie et al., 2009**). It involves a tuning parameter λ to impose a so-called $l_1$-penalty (**Tibshirani, 2011**) on the regression model. We used 4-fold cross-validation for selecting the best model, that is, the optimal value of λ (**James et al.,**

2013). In cross-validation, the set of observations is randomly divided into $k$ groups of approximately equal size. The first group is treated as a validation set, and the model is fit on the remaining $k − 1$ groups. The mean squared error is computed on the observations in the validation set, and this procedure is repeated $k$ times, each time selecting a different group of observations as the validation set. In this approach, first the $\lambda$ value corresponding to the smallest cross-validation error is selected, and then the regression model is fit to all of the available observations using the selected value of $\lambda$. Compatible with the dominance analysis presented above, the model selected by the Lasso method contained the predictors $DL_{elev}$, $TFS_{th}$, and $Int_{Flanker}$ (see *Table 1*), and the highest regression coefficients were estimated for $DL_{elev}$ and $TFS_{th}$. In addition, the Lasso procedure selected the predictor $SS_{Pcorr}$ (see *Table 1*). Thus, unlike the ordinary least-squares (OLS) multiple regression, the Lasso indicated a contribution of working memory span to speech identification in noise.

Finally, it is interesting to compare these multiple regression results to the pairwise partial correlation coefficients with the SRS, controlling for age (see *Table 2*). The predictors $DL_{elev}$ and $TFS_{th}$ showed a significantly negative partial correlation with the SRS, compatible with the results from ordinary least-squares (OLS) regression and the Lasso. As for the Lasso, the partial correlation coefficient for $SS_{Pcorr}$ was also significant. It appears possible that the (moderate) correlations between $SS_{Pcorr}$ and $DL_{elev}$ and age (see *Table 2*) increased the standard error of the regression coefficient for $SS_{Pcorr}$ in the multiple regression analysis shown in *Table 1*. The opposite pattern occurred for $Int_{Flanker}$. Here, the partial correlation coefficient for $Int_{Flanker}$ was not significant, unlike in the multiple regressions.

The relation between the scores on the SSQ questionnaire (*Gatehouse and Noble, 2004*), representing self-reported hearing abilities in daily life, and the performance in the cocktail-party listening task was analyzed via linear multiple regression. The SRS was the criterion variable, and age and the three SSQ subscales (speech hearing: $SSQ_{speech}$; spatial hearing: $SSQ_{spatial}$; other qualities: $SSQ_{qualities}$) were entered as predictors. Using the same criteria as for the regression analysis presented in *Table 1* (see Materials and methods), three participants were excluded as outliers. The model explained only a small, non-significant portion of the variance, $R^2 = 0.137$, $p=0.17$, $N = 47$. Only the regression coefficient for the 'Other qualities' scale was significant, showing a positive relation between this SSQ subscore and the SRS (see *Table 3*). Thus, persons reporting better hearing abilities on the SSQ 'Other qualities' scale tended to perform better in the spatial listening task. As seen in *Table 2*, the partial correlations controlling for age indicated a significant negative relation between the DL-elevation and $SSQ_{speech}$, and between the IPD threshold in the TFS-LF test and $SSQ_{spatial}$ and $SSQ_{qualities}$.

## Discussion

In a relatively large sample of young, normal-hearing participants ($N = 50$; age range 18–30 years), we studied the role of perceptual and cognitive factors for speech understanding in a cocktail-party situation with spatially separated interfering speakers. Our main hypothesis was that individual differences in the ability to direct auditory selective attention to the relevant stimulus, while ignoring distractors, explain a significant proportion of the inter-individual variance in cocktail-party listening performance. To test this hypothesis, we included tasks that assessed auditory and visual selective

**Table 3.** Multiple regression analysis of the relation between the SSQ scores (*Gatehouse and Noble, 2004*) representing self-reported hearing abilities (predictors) and the speech recognition score in the simulated cocktail-party listening task (criterion). $N = 47$.

| Predictor | β | SE | t | p |
|---|---|---|---|---|
| Intercept | −1.150 | 0.988 | 1.160 | 0.251 |
| Age | −0.006 | 0.034 | 0.170 | 0.864 |
| $SSQ_{speech}$ | 0.006 | 0.116 | 0.050 | 0.958 |
| $SSQ_{spatial}$ | −0.160 | 0.117 | 1.360 | 0.181 |
| $SSQ_{qualities}$ | 0.321 | 0.142 | 2.250 | 0.030 |

attention in the presence of distractors, using non-speech stimuli. Among the many different aspects of attention (cf. *Styles, 2006*), this ability seems particularly relevant in a cocktail-party situation where it is necessary to selectively attend to the target speaker and to ignore the interfering speakers and other background noise (*Bronkhorst, 2015*; *Shinn-Cunningham, 2008*).

In our experiment, the individual ability to selectively attend to an auditory target stimulus in the presence of distractors (measured in an intensity discrimination task under backward masking; *Oberfeld et al., 2014*) as well as the ability to attend to a visual target stimulus (measured in a Flanker task; *Kramer and Jacobson, 1991*) explained a significant portion of the variance in sentence identification performance with two interfering talkers. Together, the two measures of selective attention explained approximately the same proportion of variance as the binaural TFS sensitivity, which was the predictor with the highest relative importance (see *Table 1*). These results are compatible with our hypothesis that not only rather basic auditory factors like spatial hearing abilities contribute to individual differences in cocktail-party listening, but that in the cognitive domain a general ability of focusing attention on a relevant target stimulus represents an additional important predictor. One of the few previous studies that measured selective attention in the presence of distractors (*Gatehouse and Akeroyd, 2008*) found that in unaided hearing-impaired listeners (mean age about 66 years) the word recognition performance in static background noise was positively related to performance in the 'Elevator counting with distraction' subtest of the Tests of Everyday Attention. However, the analysis did not control for effects of age. In a recent study (*Cahana-Amitay et al., 2016*), a composite score for 'inhibition' that included the error rate in an incongruent condition of the Stroop color-word test (*Stroop, 1935*) was negatively correlated with sentence-final word recognition (controlling for age) in a group of subjects ranging in age between 55 and 84 years.

Our data also illustrate that it is only of limited value to use a broad, unspecific concept of 'attention' and to study associations between speech-in-noise understanding and aspects of attention that are not necessarily related to the requirements of cocktail-party listening. Thus, it seems more appropriate to recognize that attention has many facets (*Styles, 2006*), to not aggregate across the performance on tasks measuring very different aspects of attention in order to define a general 'attention' factor, and to formulate hypotheses concerning the potential importance of a particular aspect of attention for speech identification in noise. In this line of thinking, we note that in the simulated cocktail-party listening task as well as in the two tasks measuring auditory and visual selective attention, the participants had a-priori knowledge of the spatial or temporal position of the target. For this reason, the direction of attention to the target could be viewed as being endogenous (top-down) rather than exogenous (bottom up) (*Posner, 1980*). Limitations in the ability to focus attention while performing these tasks should thus be related to the dorsal fronto-parietal attentional system in the cortex rather than to the ventral network (cf. *Buschman and Miller, 2007*; *Corbetta and Shulman, 2002*). To test this hypothesis, future experiments on attentional factors influencing cocktail-party listening could include tasks measuring both the endogenous and exogenous orienting of attention, using for instance temporal or spatial cueing (e.g., *Coull and Nobre, 1998*; *Posner, 1980*). Alternatively, one could argue that in the intensity discrimination task, the onset of the backward masker elicits a capture of attention away from the target tones (e.g., *Desimone and Duncan, 1995*; *Jonides and Yantis, 1988*; *Yantis and Jonides, 1990*). Thus, it might even be necessary to further qualify the description of the particular aspect of attention that is indexed by the intensity discrimination task under backward masking and say that is measures the ability to suppress salient, but task-irrelevant auditory events.

The working memory span did not show a strong relation to cocktail-party listening. Thus, although the sentence identification task we presented requires working memory for storing the sequence of five words, the performance on this task appeared to be more strongly limited by failures of selective attention than by memory aspects. This result is compatible with previous data indicating that working memory capacity and speech identification in noise are associated in older, hearing-impaired participants (see *Akeroyd, 2008*), while in normal hearing subjects this correlation is weaker or even absent (*Füllgrabe and Rosen, 2016*). Thus, measures of selective attention should be included in future studies instead of focusing only on working memory capacity. In general, working memory and attention are not independent (*Cowan et al., 2005*). For instance, *Conway et al. (2001)* studied the probability that in a dichotic listening task participants recognize their own name on the ignored channel (*Moray, 1959*), and found that this probability was higher in participants

with a low WM span. Also, the working memory load affects speech understanding in a cocktail-party setting (*Francis, 2010*). In line with these results, in our data the working memory span showed a marginally significant negative correlation (controlling for age) with the DL-elevation (auditory selective attention), see *Table 2*.

Our results confirm the association between binaural TFS sensitivity for speech identification in a spatial listening task, compatible with earlier studies (*Füllgrabe et al., 2014*; *Neher et al., 2011*; *Neher et al., 2012*; *Ruggles et al., 2011*; *Schoof and Rosen, 2014*). The significant negative association between the IPD threshold and the SRS could be attributed to a reduced benefit from ITD cues in listeners with impaired binaural TFS sensitivity, although a reduction in TFS sensitivity could be also associated with other perceptual impairments beyond sound source localization (*Moore, 2008*). Spatial cues facilitate auditory streaming/grouping (e.g., *Darwin and Hukin, 1999*; *David et al., 2015*), selective attention (e.g., *Ihlefeld and Shinn-Cunningham, 2008*), and speech recognition (e.g., *Culling et al., 2004*). The importance of spatial cues was reported to increase with the number of interfering sound sources (*Yost et al., 1996*). Thus, one should expect a smaller influence of TFS sensitivity in situations with only one interfering speaker (compared to two as in the present study). Also, the simulated anechoic environment might have caused an overestimation of the importance of ITD cues. In a typical reverberant environment, spatial cues to sound source segregation are reduced (e.g., *Lavandier and Culling, 2010*). It remains to be shown whether the ability to attend to a target in the presence of distractors plays a stronger role than binaural TFS sensitivity in a reverberant setting. On the other hand, the spatial separation of 25° between target and distractor speakers was larger than in some previous studies that used a separation of only 15° (*Ruggles et al., 2011*, *2012*, *Ruggles and Shinn-Cunningham, 2011*). The corresponding stronger ITD cues in our study might have reduced the importance of the sensory representation of the acoustic stimulus for task performance, which would emphasize the relative importance of central factors. Note that in cocktail-party situations in daily life, the spatial separation between the target speakers and competing speakers will often be even larger than 25°.

The self-report measures of hearing abilities (SSQ scores) showed only weak associations with performance in the cocktail-party listening task (controlling for age). In hearing-impaired listeners, some earlier studies (*Gatehouse and Akeroyd, 2008*; *Heinrich et al., 2015*) also reported rather small correlations between SSQ scores and speech identification in static or amplitude-modulated background noise, even without controlling for age. *Füllgrabe et al. (2014)* found no correlation between SSQ scores and speech-in-noise perception in audiometrically normal-hearing listeners.

Compared to previous studies on predictors of cocktail-party listening, our experiments introduced several methodological improvements. The sample size in our study was larger than in some previous experiments, so that it was possible to measure the influence of several variables on sentence identification in noise, without first combining predictors into a small number of factors. We also used multiple linear regression analyses, rather than pairwise correlations as in some previous experiments. Predictors like hearing thresholds, age, working memory span and TFS sensitivity are partly correlated (see *Table 2*). Multiple linear regression accounts for correlations among the predictors (*Gauss, 1821*), while interpreting pairwise correlations is extremely difficult for a large set of intercorrelated predictors. Unlike most previous studies, we also report the reliability of the measured variables, which in most cases was acceptable to high. Finally, we used two established approaches for assessing the relative importance of predictors, dominance analysis (*Budescu, 1993*) and Lasso (*Tibshirani, 1996*). The Lasso method was proposed to avoid some of the problems of forward or backward selection in stepwise regression (*Harrell, 2015*; *James et al., 2013*).

Despite the new findings concerning the role of the ability to direct selective attention to a target stimulus for speech identification in noise and the methodological features of our study, there are of course several limitations. First, we included only a single measure each for auditory and selective attention in the presence of distractors. From a psychometric point of view, it would be desirable to include different paradigms, in order to test whether the results generalize to different tasks indexing selective attention, and to increase the reliability of the measures of selective attention. In the auditory modality, we used intensity discrimination under backward masking as an index of selective attention. In this task, the target sounds have to be selected on the basis of their temporal position within a trial. We had decided against a task measuring the *spatial* direction of attention (e.g., *Sach et al., 2000*; *Spence and Driver, 1994*) because the performance on such a task depends on abilities of spatial hearing, which we assessed separately in terms of the binaural TFS sensitivity.

Also, the precise perception of the temporal structure of speech is important for intelligibility (*Zion Golumbic et al., 2012*), and non-simultaneous masking can negatively affect speech identification (*Dirks and Bower, 1970*). Still, it would be interesting to study whether the spatial direction of attention shows a similar relation to cocktail-party listening as the temporal direction of attention. Spatial and temporal attention were reported to involve different brain areas (e.g., *Michalka et al., 2015*).

Second, we studied a relatively young group of listeners. It remains to be shown whether attentional abilities play a similar role in normal-hearing older subjects, or in hearing-impaired listeners. Age-related changes in selective attention have been reported (*Zanto and Gazzaley, 2014*), just as for other cognitive skills (e.g., *Salthouse, 1996*; *Sander et al., 2012*). At the same time, the probability of audiometrically relevant hearing losses as well as of 'hidden hearing losses' (*Plack et al., 2014*) increases with age, partly due to noise exposure across the life span. In fact, the TFS sensitivity shows a gradual deterioration with age in normal-hearing listeners (*Füllgrabe, 2013*; *Grose and Mamo, 2010*; *King et al., 2014*; *Ross et al., 2007*). Thus, the relative importance of psychoacoustic and cognitive predictors of speech understanding in a cocktail-party situation might differ between young, middle-aged and older groups, and future research should address this aspect.

Third, our participants were rather homogeneous in terms of education and (being university students) very likely also in terms of socioeconomic status and cognitive aptitude. Thus, it would be desirable to study potential psychoacoustic and cognitive predictors in a less homogeneous and more representative sample.

Fourth, our simulated cocktail-party listening task presented two spatially separated interfering speakers with the same voice as the target speaker, producing sentences with a fixed syntactical structure, and presented with a spatial separation of only 25° from the target speaker. These characteristics likely rendered the task more difficult than a realistic cocktail-party listening situation. On the other hand, communication situations in daily life often include some relatively static background noise in addition to competing speakers, while in our experiment no background noise was presented. Taken together, we assume realistic communication situations to be somewhat, but not dramatically, less difficult than the simulated cocktail-party listening task we presented. For this reason, although the average speech identification performance should be somewhat better in many realistic situations, we do not assume the latter situations to be so easy that individual differences are strongly reduced. Thus, the association between for selective attention in the presence of distractors or binaural TFS-sensitivity and speech-in-noise identification should apply to other cocktail-party situations, although ultimately this is an empirical question that should be tested in future experiments.

In conclusion, the individual ability to focus attention on a target stimulus in the presence of distractors explained a significant portion of the inter-individual variance in cocktail-party listening performance in a relatively young sample of normal-hearing listeners. Previous studies had reported that speech identification in multitalker situations is associated not only to auditory abilities such as binaural TFS sensitivity, but also to cognitive factors, predominantly in older and often hearing-impaired listeners (e.g., *Akeroyd, 2008*) but also in young normal-hearing listeners (*Zekveld et al., 2013*). Our results highlight the importance of studying aspects of attention directly relevant for speech identification in noise, rather than measuring associations with less relevant facets of attention such as visual search or to aggregate across very different aspects of attention in order to define a general 'attention' factor.

## Materials and methods

### Participants

Fifty listeners with normal hearing participated in the experiment voluntarily. All listeners reported normal hearing and no history of hearing disorders, and normal or corrected-to-normal visually acuity. They had hearing thresholds better than 20 dB HL at octave frequencies between 125 Hz and 4 kHz (that is, in the frequency region most important for speech; *Byrne et al., 1994*), calculated on the basis of equivalent threshold sound pressure levels for the Sennheiser HDA 200 earphones (*Han and Poulsen, 1998*). The maximal asymmetry between left and right ear was 15 dB in the frequency range between 125 Hz and 4 kHz. In the high-frequency range, for all but one listener the hearing thresholds were also better than 20 dB HL at 6 and 8 kHz.

The sample size of $N = 50$ was selected so that in the multiple regression analysis containing nine predictors the power to detect a moderate deviation of a single linear regression coefficient from 0 (partial $R^2 = 0.15$) was $1 - \beta = 0.8$ (two-tailed test), with $\alpha$ set to 0.05. According to G*Power (*Faul et al., 2009*), the required minimum sample size is 47.

All participants were native speakers of German. Most of them were psychology students at the Johannes Gutenberg – Universität Mainz, they received partial course credit or were paid for their participation. The experiment was conducted according to the principles expressed in the Declaration of Helsinki. All listeners participated voluntarily after providing informed written consent, after the topic of the study and potential risks had been explained to them. They were uninformed about the experimental hypotheses. The study was approved by the ethics committee of the Department of Psychology, Johannes Gutenberg-Universität Mainz.

The participants (39 female, 11 male) ranged in age between 18 and 30 years (mean age 21.5 y, SD = 3.1 y). All held the German general qualification for university entrance (Abitur), and 44 of them were psychology students. Since very good grades in secondary school are required for admission in psychology at German universities, the group can be assumed to have relatively high test intelligence, although we did not conduct an intelligence test (*Füllgrabe et al., 2014*).

## Apparatus

The auditory stimuli were generated digitally, played back via an RME (Haimhausen, Germany) ADI/S digital-to-analog converter ($f_s$ = 44.1 kHz, 24-bit resolution), attenuated by a TDT (Alachua, FL) PA5 programmable attenuator, buffered by a TDT HB7 headphone buffer, and presented via Sennheiser (Wedemark, Germany) HDA 200 circumaural headphones calibrated according to *IEC 318 (1970)*. The visual stimuli and task instructions were presented on a 17'' TFT computer monitor. The experiment was conducted in a double-walled sound-insulated chamber (IAC Acoustics Germany, Niederkrüchten). Responses were collected via a numeric keypad, a computer keyboard, or a mouse, depending on the task.

## Tasks

### Audiometric thresholds

Detection thresholds were measured bilaterally using Békésy tracking (*Békésy, 1947*; *Hartmann, 2005*) with pulsed 270-ms pure tones including 10-ms cos$^2$ on- and off-ramps. The starting frequency was 100 Hz. The frequency increased exponentially from tone to tone, at a rate of 1.4 octaves/minute. For each listener and ear, thresholds were computed as the average sound pressure level in a third-octave band around octave frequencies between 125 Hz and 4 kHz. The average hearing levels are shown in *Figure 1*. The individual better-ear pure tone average threshold (PTA$_{BE}$)

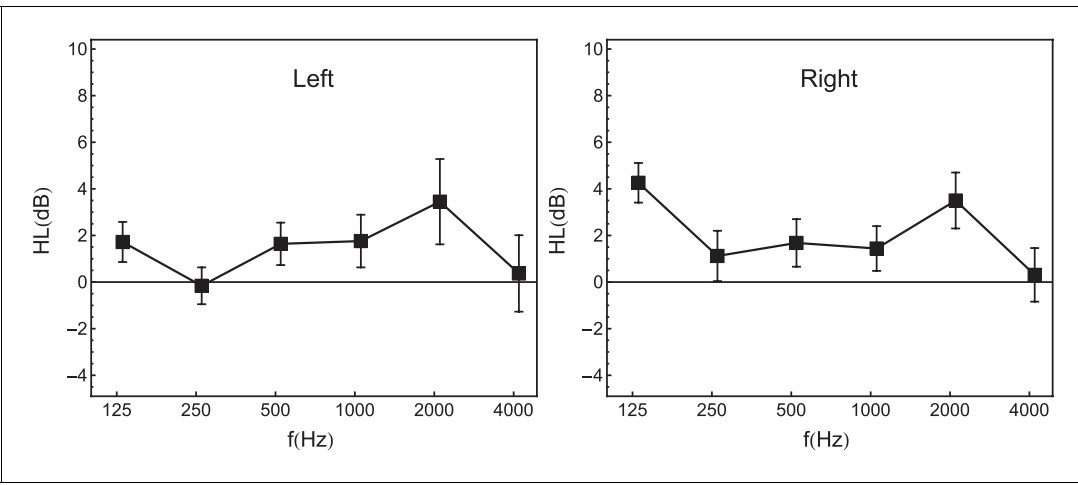

**Figure 1.** Average audiometric hearing thresholds (in dB HL), at octave frequencies between 125 Hz and 4 kHz ($N = 50$). Left panel: left ear. Right panel: right ear. Error bars represent 95% confidence intervals.

at octave frequencies between 125 Hz and 4 kHz was entered as a predictor of cocktail-party listening in the regression analyses. In addition, the individual average bilateral asymmetry of the thresholds at the same octave frequencies ($HL_{diff}$) was included as a predictor, because asymmetric thresholds can affect binaural unmasking on the basis of ITD cues (*Bronkhorst and Plomp, 1989*).

## Cocktail-party listening task

As a measure of speech understanding in adverse listening conditions, we used a simulated cocktail-party listening task with two interfering speakers. The listeners performed a sentence identification task with the speech material of the Oldenburger Satztest (OLSA; HörTech gGmbH, Oldenburg), which is a German matrix test (*Wagener et al., 1999a*). The speech material consists of sentences with the syntactic structure *name-verb-numeral-adjective-object* (e.g., 'Peter kauft vier kleine Messer' – 'Peter buys four small knives'). The sentences were constructed by pseudo-randomly selecting one of ten alternatives for each word position. This results in syntactically correct but semantically unpredictable sentences, which makes it possible to use each sentence several times for the same listener. In total, 100 different sentences are available in the OLSA test. The sentences are produced by an adult male speaker and are optimized for similar intelligibility (*Wagener et al., 1999b*). The task was to identify the sentence produced by the target speaker. The matrix of 5 (word position) × 10 (alternatives) words constituting the sentence test was displayed on a computer monitor. On each trial, subjects were asked to select the five words they had just heard using a computer mouse. The selected words were displayed in a row below the matrix of test words. Initially, the selected words were displayed in black ink. After confirming their selection by clicking on an 'Accept' button, the participants received immediate feedback concerning the correctness of their selection of words. Correctly identified words were colored in green, and incorrect words were colored in red. This visual feedback was presented for 500 ms. The next trial then started automatically after a pause of 500 ms.

The target speaker and the two interfering speakers were presented binaurally via headphones, using head related impulse responses (HRIRs) to simulate the different spatial position of the sound sources. The target speaker was presented from the front (0° azimuthal angle). The interfering speakers were presented 25° to the left and 25° to the right of the target speaker. HRIRs from an anechoic room were used because a previous study showed higher inter-individual differences in speech understanding in an anechoic condition, compared to conditions with reverberation (*Ruggles and Shinn-Cunningham, 2011*). They had been recorded with a head-and-torso simulator Brüel & Kjær Type 4128C at a distance of 80 cm between loudspeaker and microphones and an elevation of 0° (*Kayser et al., 2009*). In the experiment, the target speaker was presented at an average sound pressure level of 58 dB SPL, while each interfering speaker was presented at 60 dB SPL.

Each participant first received five trials without interfering speakers, to become familiar with the task and the response interface. Next, five trials were presented with a single interfering speaker, positioned 25° to the right of the listener. After these brief practice blocks, each listener received three experimental blocks with two interfering speakers (25° left and right), containing 50 trials each. On each trial, the sentences produced by the target speaker and the two interfering speakers were selected at random from the set of 100 test sentences, of course with the restriction that none of the three speakers produced an identical word. Note that the same male voice was used for the target speaker and the two interfering speakers, which made the task relatively difficult (*Cherry, 1953*).

For each listener and each block of 50 trials collected in the sentence identification task with two interfering speakers, the proportion of correctly identified words for the target speaker was computed (speech recognition score; SRS). Because non-normally distributed measures can cause problems in regression/correlation analyses (e.g., *Bishara and Hittner, 2012*) and repeated-measures ANOVAs (e.g., *Oberfeld and Franke, 2013*), the proportions were arcsin-square-root transformed (*Bartlett, 1936*) to obtain a closer approximation to the normal distribution. The data were analyzed with a repeated-measures analysis of variance (rmANOVA), using the multivariate approach. Partial $\eta^2$ is reported as a measure of association strength. The same type of rmANOVAs is used in all following analyses. An rmANOVA showed a significant effect of block, $F(2, 48) = 47.34$, $p<0.001$. The mean proportion of correct responses was considerably lower in the first block than in the two following blocks, compatible with data by *Wagener et al. (1999c)* who reported a sizeable practice effect in steady background noise. For this reason, the data from the first block were excluded from

further analyses. An rmANOVA conducted on the data from blocks 2 and 3 still showed a significant while rather weak effect of block on the SRS, $F(1, 49) = 4.54$, $p=0.038$, **Cohen (1988)** $d_z = 0.30$. The degree of agreement between the two measurements of the SRS (blocks 2 and 3) represents test-retest reliability and was assessed by an absolute agreement definition of the intraclass correlation in a two-way mixed-model (ICC(A,2) in the nomenclature of **McGraw and Wong, 1996**). The reliability was high, ICC(A,2) =0.934.

*Figure 2* shows a histogram of the average individual speech recognition score (proportion correct) in the simulated cocktail-party listening task (blocks 2 and 3). As expected, the listeners showed considerable variation in the SRS. The arcsin-sqrt transformed average individual proportion correct on blocks 2 and 3 (SRS) served as the measure of cocktail-party listening, and was used as the *criterion variable* in the regression analyses.

## Auditory intensity discrimination under backward masking (auditory selective attention)

Intensity difference limens (DLs) in quiet and under backward masking were measured using a two-interval, two alternative forced-choice task and an adaptive procedure with a three down, one up rule (**Levitt, 1971**). The targets and the maskers were 1-kHz pure tones with a steady-state duration of 20 ms, presented to the right ear. The tones were gated on and off with 5-ms cosine-squared ramps. The standard level was 60 dB SPL. An intensity increment – that is, a pure tone of the same frequency, duration and temporal envelope – was added in-phase to the standard in one of the observation intervals (selected randomly). The level of the backward masker was 90 dB SPL. The silent interval between standard offset and masker onset was 50 ms (see *Figure 3*). This ISI value is

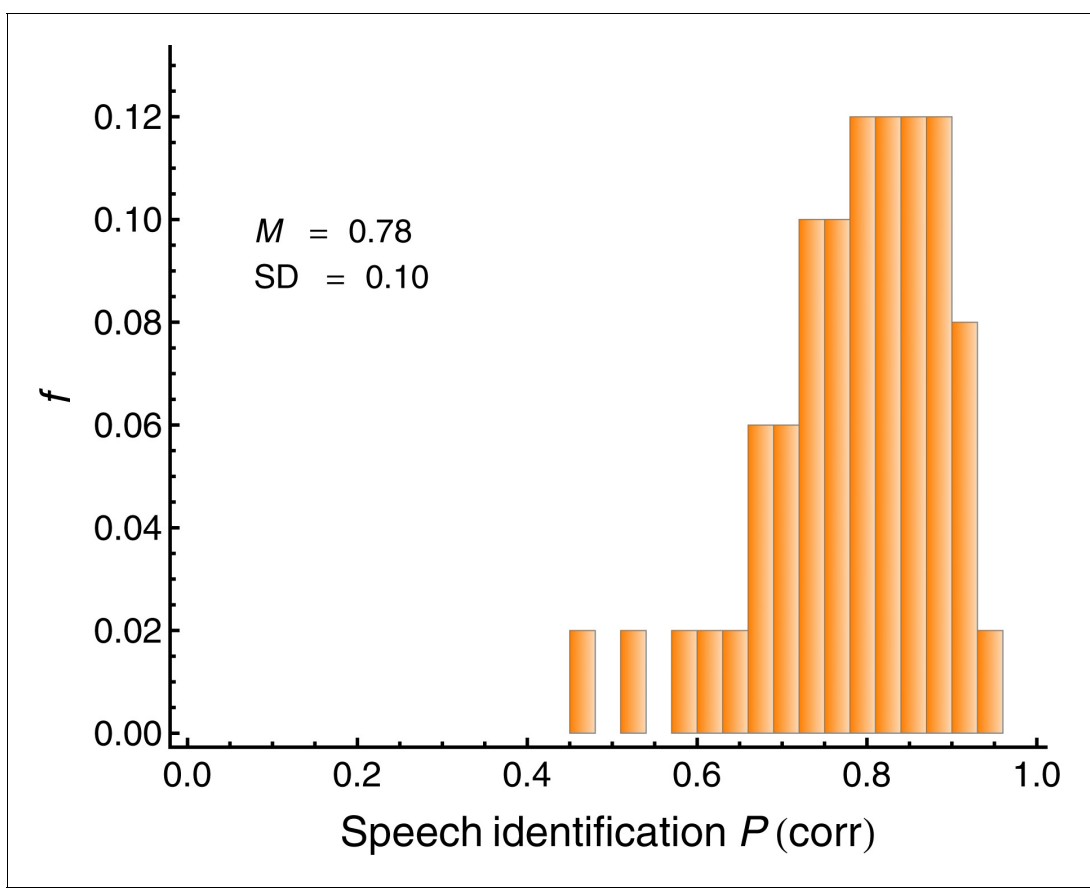

**Figure 2.** Average individual proportion correct (speech recognition score; SRS) in the simulated cocktail-party listening task with two spatially separated interfering speakers (*N* = 50). This measure served as the criterion variable in the regression analyses. The mean (*M*) and the standard deviation (SD) are displayed.

in the range where the effects of backward and forward masking on speech identification were observed (*Dirks and Bower, 1970*). The temporal interval between the onsets of the two target tones (standard and standard-plus-increment) was 800 ms. The task was to select the interval containing the louder target tone (that is, the standard-plus-increment), and to ignore the maskers. Visual trial-by-trial feedback was provided. In the adaptive procedure, the initial level of the intensity increment, expressed in terms of $10 \log_{10}(\triangle I/I)$, where $\triangle I$ is the intensity difference between the standard-plus-increment and the standard and $I$ is the intensity of the standard, was 8 dB. For the in-quiet condition, the step size was 5 dB until the third reversal, and 2 dB for the remaining six reversals. In the backward-masking condition, four reversals were collected with the larger and eight reversals with the smaller step size. The arithmetic mean of $10 \log_{10}(\triangle I/I)$ from the fourth (in quiet) or fifth reversal (backward masking) up to the last even-numbered reversal was taken as the difference limen corresponding to 79.4% correct. Adaptive tracks where the standard deviation of $10 \log_{10}(\triangle I/I)$ at the counting reversals exceeded 7 dB were excluded from the data analysis, which affected 5 tracks (1% of the total of 470 tracks). After a brief practice block, two blocks were obtained in quiet, followed by three blocks under backward masking.

An rmANOVA with the within-subjects factor block (1, 2, 3) showed no significant effect of block on the DL under backward masking, $F(2, 44) = 1.03$, $p=0.90$. Thus, there was no significant practice effect, and therefore the average individual DL under backward masking ($DL_{masked}$) across the three blocks was computed. The reliability of the masked DL across the three measurements (blocks) was moderate, ICC(A,3) = 0.871.

The average DL in quiet ($DL_{quiet}$) was included as a predictor in the regression analyses, representing a suprathreshold measure of hearing ability that is not related to selective attention. The reliability of $DL_{quiet}$ across the two measurements (blocks) was ICC(A,2) = 0.660.

We used the elevation of the intensity-DL caused by the backward masker as a measure of auditory selective attention, as in previous studies (*Oberfeld et al., 2014*). The DL-elevation denotes the difference between the DL under masking and the DL in quiet, $DL_{elev} = DL_{masked} - DL_{quiet}$. As *Figure 4* shows, there was considerable variation in the individual DL-elevations under masking, as expected.

## Binaural sensitivity for the temporal fine structure

The binaural sensitivity to temporal fine structure information was measured as the smallest detectable interaural phase difference (IPD) of a sinusoidal carrier relative to an IPD of 0°, using the TFS-LF test proposed by *Hopkins and Moore (2010)*. In a two-interval task, four pure tones (500 Hz) were presented binaurally in each interval. In one of the intervals (selected randomly), the second and fourth tone were presented with an IPD greater than 0° between the right and left ear, while the IPD

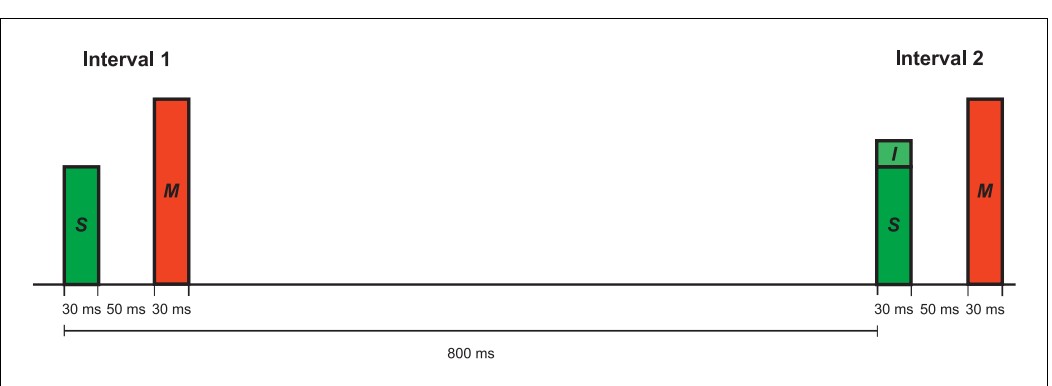

**Figure 3.** Schematic depiction of the two-interval intensity discrimination task used to measure auditory selective attention. Green: target tones. Red: backward maskers ('distractors'). The standard (*S*) was a 1 kHz tone presented at 60 dB SPL. An intensity increment (*I*) was presented in either the first or the second interval, with equal a-priori probability. The task was to select the interval containing the louder target (that is, standard-plus-increment). The maskers were 1 kHz tones presented at 90 dB SPL. The same temporal configuration was used in the in-quiet condition, except that the maskers were not presented.

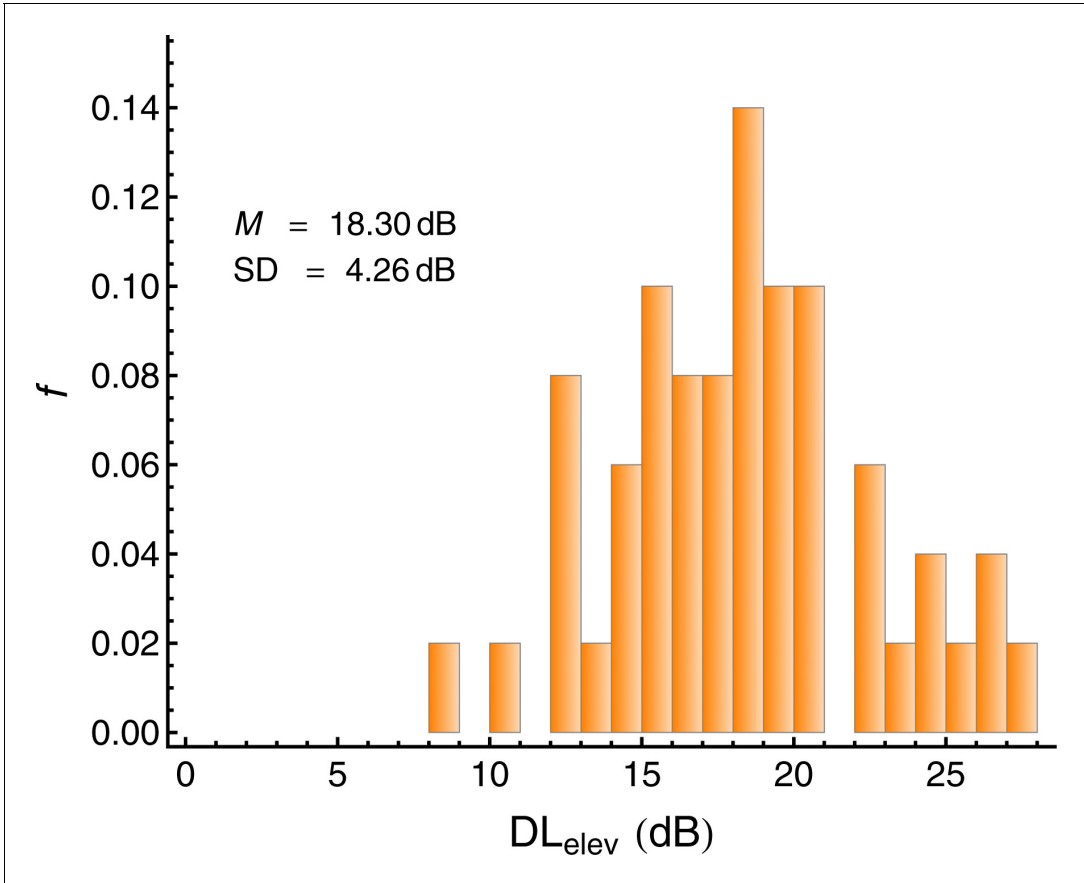

**Figure 4.** Individual elevation of the intensity difference limen caused by the backward maskers ($DL_{elev}$), defined as the difference between the DL under masking and the DL in quiet. Lower values represent a better capability of directing auditory selective attention to the target tones. $N = 50$.

was 0° for the first and the third tone. If the listener is sensitive to the change in IPD, then the four tones are perceived as changing in lateralization. In the other interval, all tones were presented with an interaural phase difference of IPD = 0°, corresponding to no change in lateralization from tone to tone. All tones were presented at 30 dB SL with a steady-state duration of 300 ms and 50 ms cosine-squared on- and offset ramps, 20 ms pauses between the tones within an interval, and 200 ms silence between the two intervals. The task was to identify the interval which contained the tones with the phase shift and thus elicited the perception of a location change. Visual trial-by-trial feedback was provided. The initial phase shift was IPD =180° and was divided by $a = (1.25)^2$ in case of three consecutive correct responses, or multiplied by $a$ after an incorrect response (three down, one up rule). After the third reversal, the step size was reduced to $a = 1.25$. The experimental block ended when nine reversals had been collected or 70 trials had been presented. The geometric mean of the IPD at the last six reversals was taken as the IPD threshold. After a brief practice block, two threshold estimates were obtained. Adaptive tracks in which the SD of the $log_{10}$- transformed values of the IPD at the counting reversals was higher than 0.3 or where less than 4 reversals had been collected were excluded from the analysis, which affected only 2 of the 100 tracks. The arithmetic mean of the IPD threshold obtained in the two blocks presenting the TFS-LF test was used as a predictor in the regression analyses ($TFS_{th}$), representing *sensitivity for the temporal fine structure*. The reliability of $TFS_{th}$ across the two measurements (blocks) was ICC(A,2) = 0.682. *Figure 5* shows the distribution of $TFS_{th}$. As expected, there was considerable inter-individual variation of the binaural TFS sensitivity, compatible with previous reports of both monaural (*Ruggles et al., 2011*) and binaural TFS sensitivity (*Füllgrabe, 2013*; *Ross et al., 2007*).

### Flanker task (visual selective attention)

To measure spatial visual selective attention, a flanker task as established by *Eriksen and Eriksen (1974)* was used, in a variant proposed by *Kramer and Jacobson (1991)*. The participants' task was to decide whether a target line presented on a computer screen was dotted or dashed. The target line was presented in vertical orientation, on the center of the display. It was surrounded by other lines, the so-called flankers. In one condition (*Figure 6*, right column), the two vertical lines adjacent to the target line were associated with the *incompatible* response. If the target line was dashed, the distractor lines were dotted, and vice versa. In a control condition (*Figure 6*, left column), the adjacent distractor lines were solid, and thus not associated with one of the responses relevant for the target line, this is the *neutral condition*. The two flanker lines adjacent to the target line were either connected with the target line with horizontal solid lines (*Figure 6*, upper row), or they were connected with two additional, vertically oriented solid lines (*Figure 6*, lower row). In the former condition ('same object'), the target line and the distractor lines can be expected to be perceived as belonging to the same visual object (*Kramer and Jacobson, 1991*). In the latter condition ('different object'), the target line and the distractors should be grouped into separate objects. According to the concept of object-based attention (e.g., *Kahneman et al., 1981*), ignoring the flankers should be more difficult if the flankers and the target are perceived as belonging to the same object. To further emphasize the grouping, the target line and the adjacent flankers were presented in the same color in the same-object condition, and in different colors in the different-object condition. The colors blue and green were used, and the target line was equally often presented in blue and in green.

The stimuli were presented on a CRT display (frame rate 85 Hz), with a viewing distance of 100 cm. The size of the vertical lines was 0.9° of visual angle (vertical) by 0.04° (horizontal). The horizontal separation between the lines was 0.25°. The trial started with a blank gray screen presented for 500

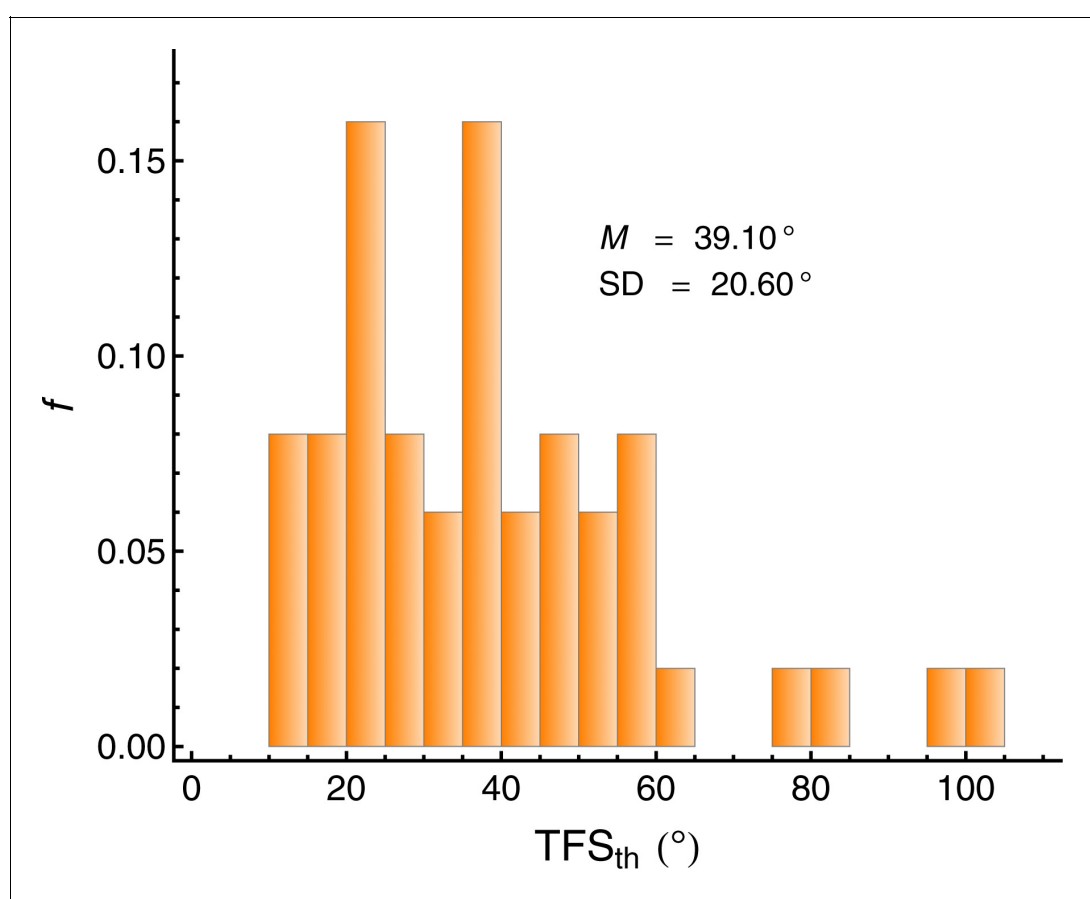

$M = 39.10°$
$SD = 20.60°$

**Figure 5.** Individual IPD thresholds in the TFS-LF test ($TFS_{th}$). Lower values represent better binaural sensitivity for the temporal fine structure. $N = 50$.

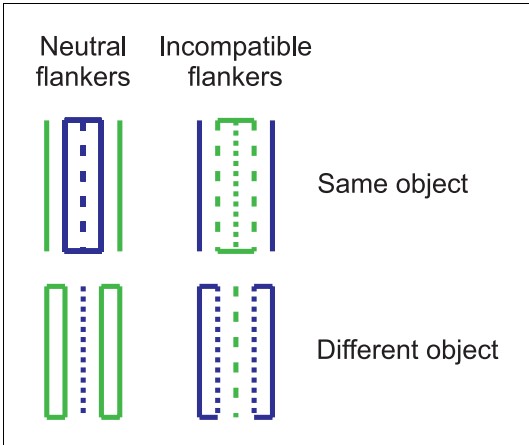

**Figure 6.** Example stimuli from the flanker task (*Kramer and Jacobson, 1991*) used to measure spatial visual selective attention. The participants' task was to decide whether the central *target line* was dotted or dashed. In the *neutral condition* (left column), the neighboring flanker lines were solid and therefore not associated with a response. In the *incompatible condition* (right column), the flanker lines were associated with the incompatible response. The horizontal lines and the colors promoted the perceptual grouping of the target line and the flankers as either belonging to the *same object* (upper row), or into *different objects* (lower row).

ms, followed by a fixation cross presented for 200 ms, after which the stimulus was presented. Participants responded by pressing two designated buttons on a numeric keypad, and received immediate visual feedback concerning the correctness of the response. They first received a practice block containing all of the 16 combinations of target type (dotted, dashed), distractor type (neutral, incompatible), object condition (same object, different object), and target color (blue, green). Then, three experimental blocks were presented. In each block, all of the 16 conditions were presented four times (64 trials/block), in random order.

Response times (RTs) below 200 ms or above 3000 ms were excluded from the analysis, which affected less than 0.1% of the trials. For each subject, the average correct RT on neutral trials ($RT_{neutral}$) was computed as a measure of *processing speed* (e.g., *Salthouse, 1996*). Because the asymmetric distribution of RTs can cause problems in regression/correlation analyses (e.g., *Bishara and Hittner, 2012*) and repeated-measures ANOVAs (e.g., *Oberfeld and Franke, 2013*), the RTs were log-transformed prior to all analyses. An rmANOVA on the RTs in the neutral condition showed marginally significant effect of block, $F(2, 98) = 2.39$, $p=0.097$. The mean RT was significantly higher in the first block than in blocks 2 and 3, representing a practice effect. For this reason, the data from block 1 were excluded from further analyses. The reliability across the two remaining blocks was high, ICC(A,2) = 0.920. The average RT on neutral trials in blocks 2 and 3 was used as a predictor ($RT_{neutral}$).

As a measure of *visual selective attention*, we used the flanker interference, defined as the difference between (log-transformed) correct RTs in the incompatible condition and the neutral condition ($Int_{Flanker} = RT_{incompatible} - RT_{neutral}$), averaged across the same-object and different-object condition and the two blocks (2 and 3). Lower values represent a better capability of directing visual selective attention to the target line (see *Figure 7*). The reliability of $Int_{Flanker}$ across the two measurements (blocks 2 and 3) was lower than desirable, ICC(A,2) = 0.596.

## Sentence span task (working memory capacity)

Working memory (WM) capacity was measured with a reading span test, originally proposed by *Daneman and Carpenter (1980)*, which is one of the most established working memory span tasks (*Conway et al., 2005*). A computer version was used (*Lewandowsky et al., 2010*). On each trial, the participants saw an alternating sequence of sentences and consonants. The task was to judge the correctness of each sentence and to remember the following consonant for later serial recall. The sentences were taken from the 'easy' variant of the German version (WMC Multilingual,

downloaded from http://www.psychologie.uzh.ch/fachrichtungen/allgpsy/Software.html). After a 1.5 s fixation cross, the first semantically correct (e.g., 'Every rabbit has fur.') or incorrect (e.g., 'Tomorrow is in the past.') sentence appeared centrally on the screen. The participants pressed one of two designated buttons on a computer keyboard to classify the sentence as correct or incorrect. On button press, the sentence disappeared and a single consonant was presented centrally for 1 s. After a 100-ms blank interval, the next sentence appeared. Depending on the list length, three to seven of these sentence-consonant sequences were presented. After the complete list had been presented, the participant was asked to type the remembered series of consonants into a response box displayed on the computer screen. The participants were required to type as many letters as were actually presented in the trial. They were informed that the order of letters mattered and were hence instructed to guess if necessary, rather than skip letters that they could not remember. No feedback was provided.

Each participant received two trials for each of the five list lengths (3, 4, 5, 6, and 7), in random order. The proportion of consonants recalled correctly (that is, reproduced in the correct list position), averaged across the 10 lists, was computed for each subject (partial credit scoring; *Conway et al., 2005*). The reliability across the two presentations of each list length was acceptable, ICC(A,2) = 0.759. The arcsin-sqrt transformed proportion correct on the sentence span task ($SS_{pcorr}$) was included as a predictor of cocktail-party listening in the regression analyses.

## Self-reported hearing problems

Self-reported hearing-related problems in daily life were assessed via the Speech, Spatial and Qualities of Hearing Scale (SSQ) by *Gatehouse and Noble (2004)*; using the German version

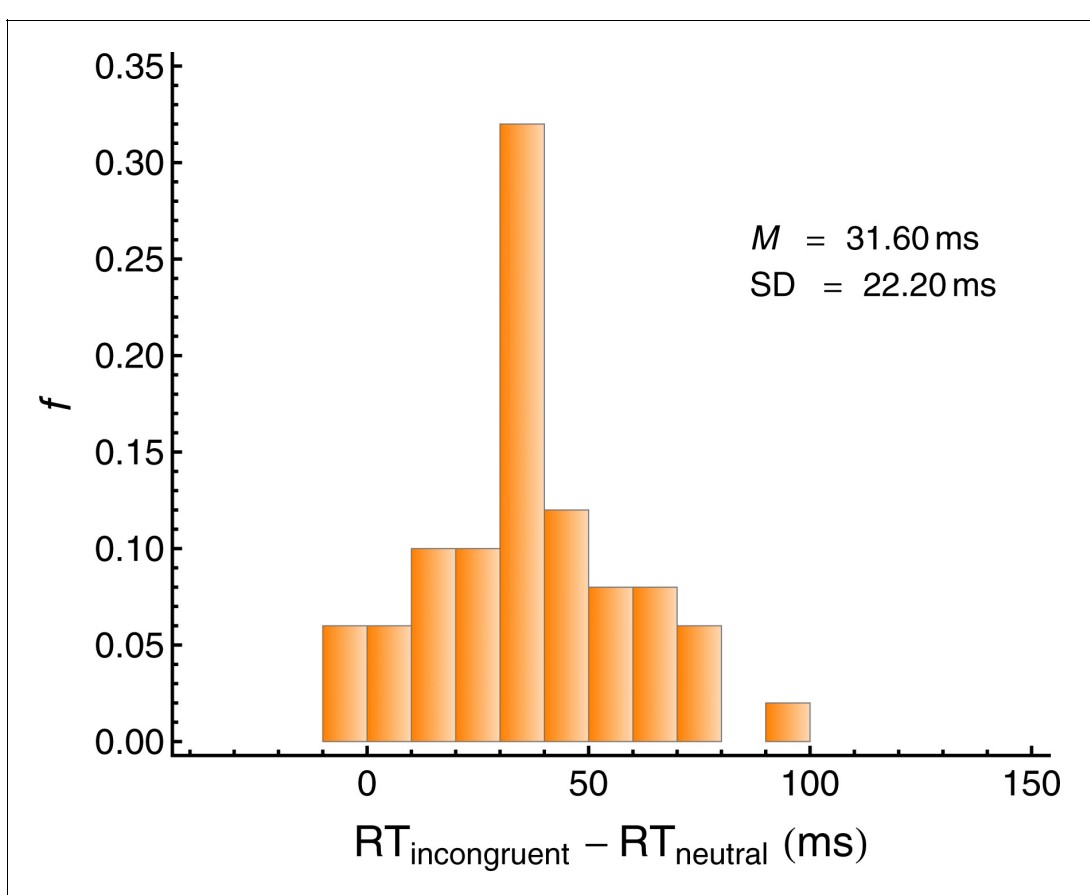

**Figure 7.** Individual flanker interference ($Int_{Flanker}$) in the visual attention task. Lower values represent a better capability of directing visual selective attention to the target line. *N* = 50.

(*Kießling et al., 2011*). The *Speech hearing* subscale covers speech understanding in the presence of additional speakers (e.g., 'You are in conversation with one person in a room where there are many other people talking. Can you follow what the person you are talking to is saying?'), and is therefore directly relevant for our research question. The *Spatial hearing* subscale indexes the capability of locating static or moving sound sources (e.g., 'You are sitting around a table or at a meeting with several people. You can't see everyone. Can you tell where any person is as soon as they start speaking?'). The *Other qualities* subscale addresses aspects of segregation of sounds, recognition, clarity/naturalness, and listening effort. The SSQ response scales range from 0 to 10, and 10 represents the highest self-rated hearing ability. The mean ratings (with SDs in parentheses) on the Speech hearing, Spatial hearing, and Other qualities scale were 7.16 (1.43), 6.95 (1.39), and 7.82 (1.18), respectively. The SSQ total score, which can range between 0 and 30, varied between 12.5 and 29.4 ($M = 21.9$, SD = 3.6). Thus, the participants showed considerable variation in their self-reported hearing abilities.

## Procedure

Each participant was tested on all tasks. To minimize inter-individual variation due to different task orders, a fixed sequence of tasks was presented. After informed written consent and basic instructions, the experiment started with the measurement of audiometric thresholds, followed by intensity discrimination in quiet, intensity discrimination under backward masking, the cocktail-party listing task, the TFS-LF test, a questionnaire concerning demographic information, the flanker task, the sentence span task, and the SSQ questionnaire. Each task was preceded by detailed instructions and practice trials. The duration of the experimental session was approximately 3 hr, including several short breaks.

## Regression analysis

Multiple linear regression was used to analyze the association between the psychoacoustic and cognitive predictors and the speech recognition score (SRS) in the cocktail-party listening task. As explained above, proportions (SRS and $SS_{Pcorr}$) were arcsin-sqrt transformed, and the response-time measures ($RT_{neutral}$, $Int_{Flanker}$) were based on log-transformed RTs. All variables were *z*-standardized. The nine predictors were entered simultaneously. Following the recommendations by *Belsley et al. (1980)*; we analyzed the externally studentized residuals, and the DFFITS index proposed by *Belsley et al. (1980)* as a measure of the influence of an observation. Observations for which the absolute value of the externally studentized residual exceeded 1.96 or with an absolute DFFITS value exceeding $2\sqrt{p/N}$ (where $N = 50$ is the number of subjects, and $p=9$ is the number of predictors) were defined as outliers. This resulted in the exclusion of 5 of the 50 subjects from the regression analysis. The maximum condition index (*Belsley et al., 1980*) was 2.49. *Belsley et al. (1980)* suggested that only condition indices of at least 30 indicate potential problems with multicollinearity. It should be noted that according to the Gauß-Markov theorem (*Gauss, 1821*) the estimates provided by the multiple regression analysis will remain unbiased in the presence of correlated predictors. However, multicollinearity could inflate the variance of the estimated regression coefficients (e.g., *Greene, 2008*), resulting in non-significant regression coefficients.

Q-Q plots of the residuals showed no systematic deviations from normality, and plots of the SRS as a function of the predictors showed no severe deviations from linearity. Thus, linear multiple regression was an appropriate method to assess the influence of the nine predictors on the speech recognition score, and to gauge their relative importance. Note that unlike most previous studies on factors influencing cocktail-party listening, our analyses did not focus on pairwise correlations, because only multiple regression provides valid information about the effects of multiple, partly correlated predictors (see *Table 2*).

Data are available from the Dryad Digital Repository (*Oberfeld and Klöckner-Nowotny, 2016*).

## Acknowledgements

This work was supported by a grant from Deutsche Forschungsgemeinschaft (www.dfg.de) to Daniel Oberfeld (OB 346/4-2: "Temporal aspects of auditory intensity processing"). The funders had no role in study design, data collection and analysis, decision to publish, or preparation of the

manuscript. The authors have no financial relationship with the organization that sponsored the research. No additional external funding received. The authors declare that they have no conflict of interest. We are grateful to Marius Frenken for his assistance in preparing the figures, and to Annika Grotjohann, Marius Frenken, Julia Pfeiff, and Jannis Renner for their help with data collection. We thank Eve Marder, Barbara Shinn-Cunningham, Hari Bharadwaj and an anonymous reviewer for helpful comments on a previous version of this paper.

## Additional information

### Funding

| Funder | Grant reference number | Author |
|---|---|---|
| Deutsche Forschungsge-meinschaft | OB 346/4-2 | Daniel Oberfeld |

The funders had no role in study design, data collection and interpretation, or the decision to submit the work for publication.

### Author contributions

DO, FK-N, Conception and design, Acquisition of data, Analysis and interpretation of data, Drafting or revising the article

### Author ORCIDs

Daniel Oberfeld, http://orcid.org/0000-0002-6710-3309

### Ethics

Human subjects: The experiment was conducted according to the principles expressed in the Declaration of Helsinki. All listeners participated voluntarily after providing informed written consent, after the topic of the study and potential risks had been explained to them. They were uninformed about the experimental hypotheses. The study was approved by the ethics committee of the Department of Psychology, Johannes Gutenberg-Universitaet Mainz.

## Additional files

### Major datasets

The following dataset was generated:

| Author(s) | Year | Dataset title | Dataset URL | Database, license, and accessibility information |
|---|---|---|---|---|
| Oberfeld D, Klöckner-Nowotny F | 2016 | Experimental data on psychoacoustic and cognitive predictors of cocktail-party listening | http://dx.doi.org/10.5061/dryad.f96cr | Available at Dryad Digital Repository under a CC0 Public Domain Dedication |

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
