## [Decision Letter]

Thank you for submitting your article "Individual differences in selective attention predict speech identification at a cocktail party" for consideration by *eLife*. Your article has been reviewed by three peer reviewers, including Barbara G Shinn-Cunningham (Reviewing editor and Reviewer #1) and Hari Bharadwaj (Reviewer #2), and the evaluation has been overseen by Eve Marder as the Senior Editor.

The reviewers have discussed the reviews with one another and the Reviewing Editor has drafted this decision to help you prepare a revised submission. We hope you will be able to submit the revised version within two months, so please let us know if you have any questions first.

Summary:

This investigation tests the hypothesis that individual differences in the ability to identify target speech in the presence of spatially separated speech distractors is related to cross-subject differences in both supra-threshold auditory temporal processing abilities and cognitive function. More specifically, the authors assess binaural sensitivity to temporal-fine-structure (TFS) information and the ability to focus attention in a large group of young audiometrically normal-hearing listeners, thereby extending and complementing previous work. Overall, this is a nice study in distinguishing between sensory and central contributions to individual differences that may affect everyday communication. The study is clear, well written, and of interest to the hearing-research community. We have three major issues that need to be addressed, along with some suggestions that might strengthen your presentation.

Essential revisions:

1) The statistical tests you performed do not disprove a sensory contribution to the observed individual differences that you argue reflect top-down attention.

You have shown that the intensity difference limen (DL) measured using backward masking correlates strongly with DL in quiet (Table 2). Given that DL_masked_ and DL_quiet_ are strongly correlated, using them both as predictors in the multiple regression analysis makes the assignment of coefficients (and subsequent t-scores) dependent on the exact algorithm used for fitting, and essentially arbitrary.

If DL_masked_ was not included, was DL_quiet_ a significant predictor? Is DL_masked_ a significant predictor when DL_quiet_ has already been partialed out?

The Lasso fitting procedure penalizes the L1-norm of the set of coefficients, thereby favoring sparse solutions. Thus, it would pick the better of the two predictors: that is, DL_masked_ might be picked over DL_quiet_ or over both together. However, the fact that the procedure does not pick DL_quiet_ is not in itself evidence that DL_masked_ does not include differences from sensory factors. All this proves is that the DL_masked_ is the better of the two predictors overall and effectively encompasses the other.

Also, you use an eigenvalue ratio criterion (First paragraph of Regression analysis section) to say that there isn't a multicollinearity problem here. This index is of limited utility when many noisy predictors are used. Indeed the correlation between DL_masked_ and DL_quiet_ is already evidence for some multi collinearity.

The language you use to describe your results needs to reflect these subtleties, and/or you need to include more extensive statistical testing to rule out sensory contributions that are correlated with/included in DL_masked_. All you can currently conclude is that the perceptual weight given to the masker accounts for the behavior, not that this factor reflects only top-down selection.

2) Isn't a differential measure of auditory masking effects more in line with your other metrics?

If the differences in DL under masking reflect top-down selection, it seems to make more sense to quantify the differential effect of masker on DL as a correlate rather than the overall threshold under masking (e.g., DL_masked_ minus DL_quiet_). This would better parallel the visual measure you use (the differential effect of the flanker on the reaction time) as a measure of top-down selection (as opposed to the overall RT with an incongruent flanker).

3) The "cognitive" component you measure seems more specific than your discussion/presentation concedes.

The backward masking task you used to isolate auditory "attention" requires listeners to attend to a signal and ignore a later event that draws attention exogenously. This task no doubt indexes the ability to suppress an exogenous draw of attention, and your results show that this ability differs across listeners in a way that is reflected in "cocktail party" listening. However, this may be very different from the ability to sustain focused attention. Indeed, different attentional control subnetworks are thought to control exogenous vs. endogenous attention; a whole host of processes (e.g., precise sensory coding, scene analysis, top-down selection, exogenous attention.) allow us to selectively understand a target and ignore maskers.

In the first paragraph of discussion, you claim that "the ability to direct auditory selective attention" differs in consistent ways across listeners; however, it seems more appropriate to argue that there are differences in "the ability to suppress exogenously salient, but task-irrelevant auditory events." See also the second paragraph, and eighth paragraph of Discussion section where you write, "future experiments on attentional factors influencing cocktail-party listening could include tasks measuring both the endogenous and exogenous orienting of attention." We agree; but we suspect you have just isolated one particular aspect of attention, already.

This does not diminish the importance of your study, but changes the language you might use to discuss your results. Please consider integrating this idea into the way you describe your study. Rather than arguing that you are measuring "attention" (and leaving it at that), taking a more sophisticated, nuanced view that distinguishes amongst different aspects of attention, and that doesn't over-simplify, would enhance the impact of your work.

---

## [Author Response]

*Essential revisions:*

*1) The statistical tests you performed do not disprove a sensory contribution to the observed individual differences that you argue reflect top-down attention.*

2) Isn't a differential measure of auditory masking effects more in line with your other metrics?

The problems that were correctly identified under point 1 are now solved in the revised manuscript by addressing point 2. For this reason, we respond to the detailed comments that were listed under point 1 and point 2 in a slightly non-chronological order.

*If the differences in DL under masking reflect top-down selection, it seems to make more sense to quantify the differential effect of masker on DL as a correlate rather than the overall threshold under masking (e.g., DL_masked_ minus DL_quiet_). This would better parallel the visual measure you use (the differential effect of the flanker on the reaction time) as a measure of top-down selection (as opposed to the overall RT with an incongruent flanker).*

We agree and are grateful for this suggestion. We now used the elevation in the intensity DL caused by the backward maskers as a measure of auditory selective attention, defined as DL_elev_ = DL_masked_ − DL_quiet_ as in several of our previous studies (e.g., Oberfeld & Stahn, 2012, PLOS One). This separates the basic sensory sensitivity (DL_quiet_) in the intensity discrimination task from the detrimental effects of the distractors/maskers.

*You have shown that the intensity difference limen (DL) measured using backward masking correlates strongly with DL in quiet (Table 2). Given that DL_masked_ and DL_quiet_ are strongly correlated, using them both as predictors in the multiple regression analysis makes the assignment of coefficients (and subsequent t-scores) dependent on the exact algorithm used for fitting, and essentially arbitrary.*

As can be seen in Table 2 of the revised paper, using the DL-elevation instead of DL_masked_ as predictor solves this problem. The partial correlation (controlling for age) between DL_elev_ and DL_quiet_ is close to 0. The same applies to the "raw" pairwise correlations (not controlling for age), as shown in the table included below. Also, DL_elev_ shows no substantial correlations with any other predictor (see Table 2 of the revised paper and the table below), except probably the sentence span measure (SS_PCorr_; ρpartial = −0.255, p =.077), which clearly represents a cognitive rather than sensory aspect.

*If DL_masked_ was not included, was DL_quiet_ a significant predictor? Is DL_masked_ a significant predictor when DL_quiet_ has already been partialed out?*

*The Lasso fitting procedure penalizes the L1-norm of the set of coefficients, thereby favoring sparse solutions. Thus, it would pick the better of the two predictors: that is, DL_masked_ might be picked over DL_quiet_ or over both together. However, the fact that the procedure does not pick DL_quiet_ is not in itself evidence that DL_masked_ does not include differences from sensory factors. All this proves is that the DL_masked_ is the better of the two predictors overall and effectively encompasses the other.*

*The language you use to describe your results needs to reflect these subtleties, and/or you need to include more extensive statistical testing to rule out sensory contributions that are correlated with/included in DL_masked_. All you can currently conclude is that the perceptual weight given to the masker accounts for the behavior, not that this factor reflects only top-down selection.*

These issues are solved by using DL_elev_ instead of DL_masked_, because DL_elev_ and DL_quiet_ are virtually uncorrelated.

*Also, you use an eigenvalue ratio criterion (First paragraph of Regression analysis section) to say that there isn't a multicollinearity problem here. This index is of limited utility when many noisy predictors are used. Indeed the correlation between DL_masked_ and DL_quiet_ is already evidence for some multi collinearity.*

We agree that there is no general consensus concerning the optimal criterion for detecting multicollinearity, although the condition index is widely used in the literature. Concerning the two measures derived from the intensity discrimination task, these issues are solved because DL_elev_ and DL_quiet_ are virtually uncorrelated. On a more general level, the Gauß- Markov theorem states that the regression coefficients are unbiased even if there are correlations between predictors. However, multicollinearity could result in non-significant regression coefficients due to an increase in the standard error of the estimated regression coefficients. We consider it unlikely that this was a serious problem in our analyses, however, because there were only a few substantial pairwise correlations between predictors (Table 2), and because the pairwise partial correlations (where multicollinearity cannot have affected the results) between the SRS and the predictors showed a significant correlation for only one predictor (SS_Pcorr_) for which the regression coefficient in the multiple regression was non-significant. On a more general level, we believe that multiple regression provides a more informative view of the associations than the pairwise partial correlations, and that correlations between predictors are not argument against but for using multiple regression. These aspects are discussed in the revised paper:

Results: "It appears possible that the (moderate) correlations between SS_Pcorr_ and DL_elev_ and age (see Table 2) increased the standard error of the regression coefficient for SS_Pcorr_ in the multiple regression analysis shown in Table 1."

Materials and methods: "The maximum condition index (Belsley et al. 1980) was 2.49. Belsley et al. (1980) suggested that only condition indices of at least 30 indicate potential problems with multicollinearity. […] However, multicollinearity could inflate the variance of the estimated regression coefficients (e.g., Greene 2008), resulting in non-significant regression coefficients."

*3) The "cognitive" component you measure seems more specific than your discussion/presentation concedes.*

*The backward masking task you used to isolate auditory "attention" requires listeners to attend to a signal and ignore a later event that draws attention exogenously. This task no doubt indexes the ability to suppress an exogenous draw of attention, and your results show that this ability differs across listeners in a way that is reflected in "cocktail party" listening. However, this may be very different from the ability to sustain focused attention. Indeed, different attentional control subnetworks are thought to control exogenous vs. endogenous attention; a whole host of processes (e.g., precise sensory coding, scene analysis, top-down selection, exogenous attention.) allow us to selectively understand a target and ignore maskers.*

*In the first paragraph of discussion, you claim that "the ability to direct auditory selective attention" differs in consistent ways across listeners; however, it seems more appropriate to argue that there are differences in "the ability to suppress exogenously salient, but task-irrelevant auditory events." See also the second paragraph, and eighth paragraph of Discussion section where you write, "future experiments on attentional factors influencing cocktail-party listening could include tasks measuring both the endogenous and exogenous orienting of attention." We agree; but we suspect you have just isolated one particular aspect of attention, already.*

*This does not diminish the importance of your study, but changes the language you might use to discuss your results. Please consider integrating this idea into the way you describe your study. Rather than arguing that you are measuring "attention" (and leaving it at that), taking a more sophisticated, nuanced view that distinguishes amongst different aspects of attention, and that doesn't over-simplify, would enhance the impact of your work.*

We agree, and this was exactly why in the manuscript we say early on that we are interested in "selectively attending to a target in the presence of distractors" rather than speaking about "attention" per se. We now double-checked the text for 5 passages where our wording was too unspecific, and now use this more specific description throughout the text. Also, the differences between endogenous and exogenous attention were already discussed in the original manuscript. However, we reached a different conclusion and believe that because the spatial position of the target speaker in the cocktail-party listening task and the temporal position of the target tones in the intensity discrimination task were fixed and known, these tasks involved an endogenous rather than exogenous direction of attention. In the revised paper, we introduce this aspect earlier in the Discussion section, discuss that capture of attention by the backward maskers is an alternative "exogenous attention" view of our task, and added the qualification that the intensity discrimination task might even more specifically address "the ability to suppress salient, but task-irrelevant auditory events.":

"To test this hypothesis, future experiments on attentional factors influencing cocktail-party listening could include tasks measuring both the endogenous and exogenous orienting of attention, using for instance temporal or spatial cueing (e.g., Coull and Nobre 1998; Posner 1980). Alternatively, one could argue that the onset of the backward masker elicits a capture of attention away from the target tones (e.g., Desimone and Duncan 1995; Jonides and Yantis 1988; Yantis and Jonides 1990). Thus, it might even be necessary to further qualify the description of the particular aspect of attention that is indexed by the intensity discrimination task under backward masking and say that is measures the ability to suppress salient, but task-irrelevant auditory events."